# Spike Formation Is a Turning Point Determining Wheat Root Microbiome Abundance, Structures and Functions

**DOI:** 10.3390/ijms222111948

**Published:** 2021-11-04

**Authors:** Alla Usyskin-Tonne, Yitzhak Hadar, Dror Minz

**Affiliations:** 1Soil, Water and Environmental Sciences, Volcani Research Center, Rishon LeZion 7505101, Israel; al.87.la@gmail.com; 2Robert H. Smith Faculty of Agriculture, Food and Environment, The Hebrew University of Jerusalem, Rehovot 7610001, Israel; yitzhak.hadar@mail.huji.ac.il

**Keywords:** wheat, spike, root-associated, rhizosphere, microbiome, metagenome

## Abstract

Root selection of their associated microbiome composition and activities is determined by the plant’s developmental stage and distance from the root. Total gene abundance, structure and functions of root-associated and rhizospheric microbiomes were studied throughout wheat growth season under field conditions. On the root surface, abundance of the well-known wheat colonizers *Proteobacteria* and *Actinobacteria* decreased and increased, respectively, during spike formation, whereas abundance of *Bacteroidetes* was independent of spike formation. Metagenomic analysis combined with functional co-occurrence networks revealed a significant impact of plant developmental stage on its microbiome during the transition from vegetative growth to spike formation. For example, gene functions related to biofilm and sensorial movement, antibiotic production and resistance and carbons and amino acids and their transporters. Genes associated with these functions were also in higher abundance in root vs. the rhizosphere microbiome. We propose that abundance of transporter-encoding genes related to carbon and amino acid, may mirror the availability and utilization of root exudates. Genes related to antibiotic resistance mechanisms were abundant during vegetative growth, while after spike formation, genes related to the biosynthesis of various antibiotics were enriched. This observation suggests that during root colonization and biofilm formation, bacteria cope with competitor’s antibiotics, whereas in the mature biofilm stage, they invest in inhibiting new colonizers. Additionally, there is higher abundance of genes related to denitrification in rhizosphere compared to root-associated microbiome during wheat growth, possibly due to competition with the plant over nitrogen in the root vicinity. We demonstrated functional and phylogenetic division in wheat root zone microbiome in both time and space: pre- and post-spike formation, and root-associated vs. rhizospheric niches. These findings shed light on the dynamics of plant–microbe and microbe–microbe interactions in the developing root zone.

## 1. Introduction

The rhizosphere is a hotspot for plant and soil microbiome interactions. Rhizosphere microbiome composition and function are mostly shaped by soil type, plant species and plant genotypes [1,2,3,4]. Plants select their associated bacteria by depositing specific exudates in the soil–root interface [5,6]. Roots secrete large amounts of carbon as exudates that can be divided into two major classes of metabolites. The first consists of high-molecular-weight compounds such as polysaccharides and proteins, and the second of low-molecular-weight metabolites, including amino acids, organic acids, ions, sugars, vitamins and various secondary metabolites [7,8,9]. The root tip is the major source of root exudation [10,11,12], along with root elongation zones and root hairs [13,14]. As a result, a gradient of exudates is created from the roots to the surrounding soil, with the highest concentration near the root surface [15]. Thus, the root surface provides a more nutrient-rich environment than the rhizosphere, enabling microorganisms to colonize and proliferate.

Plant roots and their microbiome (both root-associated and rhizosphere) are continuously interacting. The root-associated bacteria can manipulate plant exudates by accelerating their diffusion, as well as altering plant carbon allocation to the roots [16]. In many plant species, specific microbiome functions have been shown to be associated with the microorganism–plant interactions. These functions include motility and adhesion to roots, biofilm formation, metabolism of nitrogen, carbohydrates and vitamins and degradation of xenobiotics [4,17,18]. In *Arabidopsis*, it has been proposed that roots of young seedlings secrete high levels of sugars to attract colonization of diverse bacteria, whereas in older plants, roots secrete phenolic substances to select specific rhizosphere bacteria [19,20]. A plant’s developmental stage has also been shown to affect root-exudation patterns [16,21]. This developmental stage-dependent exudation was recently shown to influence rhizosphere community structure in broomcorn millet [22] and root-associated community structure in rice [23]. 

Although the root surface provides an environment that is much richer in nutrients, most studies have focused on the influence of exudates on the rhizospheric bacterial community [1,9,24,25,26] rather than the root-associated one. Moreover, the effects of plant growth stage on rhizosphere community functions, and particularly on the root-associated community, remain unclear. To close this gap, we followed field-grown wheat throughout the growth season from emergence to spike maturation and compared the rhizosphere bacterial community (inhabiting the soil surrounding the roots) to that of root-associated bacteria (sampled from washed roots). We describe and discuss the changes in root-associated and rhizosphere bacterial community structure and functions with stages of plant development. 

## 2. Results

Wheat grown under commercial field conditions was sampled from sprout emergence up to week 17, when some plants produced green spikes whereas in others, the spikes were already more mature. Rhizosphere (soil sampled by shaking from the roots) and washed roots (root-associated) were collected at weeks 1, 2, 5, 9, 12 and 17. The spatial (rhizosphere vs. root-associated) and temporal (before *vs*. after spike formation) effects on the bacterial community structures and functional profiles were analyzed. 

### 2.1. Influence of Niche and Plant Developmental Stage on Bacterial Community Structure and Functional Profile

After filtering out the plant host genes from those of the root and rhizosphere microbiomes, around 80% of the sequences were related to bacteria, 16% could not be assigned to any known taxonomic group, and the rest were assigned to Eukaryota and Archaea. The Eukaryota, Archaea, and unassigned sequences were filtered out from the root and rhizosphere microbiome and the bacterial sequences were further analyzed; 28% and 20% (root and rhizosphere, respectively) of those sequences were assigned to known KEGG functions.

Clustering patterns of the bacterial community structure (Figure 1A) and gene functions (Figure 1B) in the two niches (i.e., root-associated and rhizosphere) are presented in nMDS orientation; nMDS orientation of bacterial community structure was based on all annotated bacterial taxonomic levels (Figure 1A). Gene functions in the nMDS orientation (Figure 1B) were constructed based on annotation of genes with known KO groups from the KEGG database. Niche origin had a major influence on both bacterial community structure and gene functions (calculated using the Adonis function in the Vegan R package; *p* < 0.0001). A lesser, but still significant influence of plant age (especially before and after spike formation) on the combined root-associated and rhizosphere bacterial community structure (*p* < 0.005) and functions was found (*p* < 0.05). The influence of wheat age was more pronounced in each niche separately than in the two niches combined. Spike formation was the turning point, leading to major shifts in both root-associated and rhizosphere bacterial community structures and gene functions. Thus, throughout the study, we compared the results obtained before and after spike formation. Differences between the niches were found in each week from sprout emergence until spike formation, in the structure of the bacterial communities (Figure 1C) and their gene functions (Figure 1D). However, niche differences for both parameters decreased after spike formation.

In addition, spike formation was also the turning point, leading to significant increase in total bacteria abundance in root-associated community (Figure 2A, e.g., 16S rRNA/*TEF* and bacteria genes per plant genes from metagenome). Surprisingly, not only spike formation influenced community total root-associated gene function but also the specific functional gene *nosZ* (Figure 2B), known as sole enzyme capable to reduce N_2_O, a greenhouse gas [27,28]. With spike formation, abundance of *nosZ* gene per total root-associated bacterial community decreased, possibly indicating diminished root-associated microbiome capability to reduce N_2_O. Unlike abundance of *nosZ* per total root bacterial community, its abundance per rhizosphere community was developmental stage dependent. In the rhizosphere, *nosZ* abundance per total community have increased with plant age (Figure 2B). This pattern can be explained by the difference in nitrate availability according to the proximity to the root.

Structural co-occurrence networks were constructed based on bacterial phylogeny of all annotated taxonomic levels (nodes) from the metagenome (for simplicity, nodes were marked with bacterial phyla). During wheat growth, the number of interactions (edges) between bacterial communities in the rhizosphere was almost eight times higher than that between root-associated bacterial communities (5019 vs. 641 edges; Figure 3A,B). When the complete network (all time points together) was divided between pre-spike formation and post-spike formation, the number of edges increased: whereas the complete network for root-associated bacterial communities was composed of 641 edges, pre- and post-spike formation networks were composed of 1207 and 2411 edges, respectively (Figure 3A). Similar trends were observed for the rhizosphere (Figure 3B). This supports the hypothesis that spike formation has a very strong influence on the microbiome, affecting both root-associated and rhizosphere bacterial communities (Figure 1A,B and Figure 2). 

Networks of functional genes co-occurrence were constructed based on annotated genes with known KO numbers from the KEGG database. These included six categories: cellular processes, environmental information processes, genetic information processes, diseases, metabolism and systems. The number of interactions in the co-occurrence networks of functional genes in the root-associated bacterial community was slightly higher compared to the rhizosphere bacterial community during wheat growth (Figure 3C,D, respectively). Five distinct functional gene clusters in the root-associated community interacted in the co-occurrence networks before spike formation, compared to only one cluster after spike formation (Figure 3C). The biggest cluster in the root-associated microbiome before spike formation was related to metabolism of carbohydrates, amino acids, nucleotides and vitamins. Another cluster was related to processes involved mainly in carbon fixation, which negatively interacted with metabolism of methane, nitrogen and secondary metabolites, and to plant–pathogen interactions. The third cluster was related to processes involved in quorum-sensing, biofilm formation and cell motility. The fourth cluster was related to processes involved in transcription and the immune response. The fifth cluster was related to processes involved in proliferation of bacterial infections, prokaryotic defense system and metabolism of secondary metabolites, including antibiotics. The last cluster found in the root-associated microbiome after spike formation was related to processes involved mainly in carbon fixation, and metabolism of methane, nitrogen and carbohydrates (Figure 3C). In the rhizosphere, on the other hand, three distinct clusters were formed after spike formation, compared to only one before spike formation (Figure 3D). The latter cluster was related to processes involving mainly in carbohydrate metabolism and transcription. The biggest cluster in the rhizosphere after spike formation was related to biosynthesis of secondary metabolites, enzymes, signal transduction, membrane transport, cell motility and amino acid and nucleotide metabolisms. The two other clusters were related to antimicrobial resistance genes, which interacted negatively with energy metabolism, and membrane trafficking, which interacted positively with transcriptional regulators.

### 2.2. Influence of Niche and Spike Formation on Bacterial Community Composition 

At the phylum level, *Proteobacteria* dominated the wheat root-associated bacterial community (47–67% relative abundance), followed by *Actinobacteria* (23–42%) and *Bacteroidetes* (4–6.5%) (Figure 4A). In the rhizosphere, *Actinobacteria* were dominant (44–50%), followed by *Proteobacteria* (32–35%) and *Acidobacteria* (9–12%) (Figure 4A). *Proteobacteria*, *Actinobacteria* and *Bacteroidetes* showed a niche preference and were dependent on plant growth stage. *Proteobacteria* and *Bacteroidetes* and their affiliated groups were more abundant in the root-associated bacterial community compared to the rhizosphere, while most *Actinobacteria*-affiliated groups were more abundant in the rhizosphere (Figure 4B). The abundance of *Proteobacteria* in the roots was significantly higher before spike formation, whereas the abundance of *Actinobacteria* was higher after spike formation (Figure 4C). The abundance of *Bacteroidetes* in the root-associated community was not affected by spike formation. In the rhizosphere, the abundance of *Acidobacteria* was higher before spike formation, whereas the abundance of *Actinobacteria* and *Proteobacteria* was higher after spike formation. 

### 2.3. The Effect of Niche and Spike Formation on Specific Root-Associated and Rhizosphere Gene Functions 

To zoom in on functional genes that were significantly enriched in one niche over the other (root or rhizosphere) or before or after spike formation, they were divided into eight functional categories to reduce data complexity. The functional categories were related to: (1) carbons; (2) nitrogen; (3) biofilm and sensorial movement; (4) antibiotic synthesis and resistance; (5) ion transporters; (6) amino acid transporters; (7) vitamin transporters; and (8) metal transporters (Figure 5A). Overall, during wheat growth, the highest enrichment in functional gene abundance in the root-associated compared to rhizosphere bacterial community was related to two categories: biofilm and sensorial movement, and antibiotic synthesis and resistance (Figure 5B).

The biofilm and sensorial movement group included genes related to secretion systems type I, III, IV and VI, secretion systems related to pili and flagella and mannose-sensitive hemagglutinin (MSHA), biofilm formation, motility, quorum-sensing and chemotaxis. Interestingly, 96 genes from this group were abundant in the root-associated community before spike formation, and none were enriched after spike formation (Figure 6A,B). Antibiotic synthesis and resistance genes were separated into two subgroups in relation to the timing of spike formation: multidrug and resistance toward (1) beta-lactams, (2) antimicrobial peptides and (3) invading RNA elements (CRISPR systems) before spike formation, and biosynthesis of various antibiotics, including polyketide, after spike formation (Figure 6B).

The gene cluster related to carbons consisted mainly of transporters and had twice the number of significantly abundant functional genes in the roots compared to the rhizosphere (136 vs. 67 genes; Figure 5B), while amino acid transporters showed almost three times the number of genes (22 vs. 8 genes, respectively). Genes involved in carbon compound biosynthesis and degradation processes were more abundant in roots than in the rhizosphere. In addition, significantly enriched genes related to amino acid transporters (possibly related to availability of root exudates) were two times more abundant before compared to after spike formation in the roots (17 vs. 9 genes) (Figure 6B,A). 

Out of the genes related to the nitrogen cycle, two gene groups (urea transporter and denitrification) showed significant niche preference. Genes related to the urea transporter are more abundant in root-associated bacterial community, while genes related to denitrification are more abundant in the rhizosphere community (Figure 5B).

In the rhizosphere, the number of significantly enriched genes assigned to one of the eight selected functional groups (i.e., related to carbons, nitrogen, biofilm and sensorial movement, antibiotic synthesis and resistance, ion transporters, amino acids, vitamins and metal transporters) was much smaller (23%) compared to root-associated bacteria (116 vs. 503 genes), respectively. The significantly enriched genes in the rhizosphere belonged to three groups: (1) carbon transporters, biosynthesis and metabolism; (2) antibiotic biosynthesis and resistance and (3) amino acid biosynthesis and metabolism (Figure 6A,C). Significantly abundant genes in the rhizosphere assigned to antibiotic biosynthesis and resistance and to amino acid biosynthesis and metabolism were more abundant before spike formation. 

It is interesting to note that only nine functions (lipopolysaccharide transporter and biosynthesis, beta-lactam resistance, metabolism of methane, metabolism of alanine, aspartate and glutamate, metabolism of arginine and proline and biosynthesis of peptidoglycan, lysine and enediyne antibiotics, Figure 6B) comprised up to 45% of the significantly enriched genes in the rhizosphere. Their abundance was significantly influenced by spike formation in both rhizosphere and roots. 

### 2.4. Links between Taxonomy and Significantly Changed Functions 

Links between selected representative bacterial phyla and significantly abundant functional genes before and after spike formation in the roots are shown in Figure 7B. The phyla *proteobacteria*, *Actinobacteria* and *Bacteroidetes* were selected for these analyses for three reasons: (1) these three groups had the highest relative abundance among all phyla in the roots (Figure 3A); (2) *Proteobacteria* and *Bacteroidetes* were significantly enriched in the roots compared to the rhizosphere; (3) *Proteobacteria* and *Actinobacteria* abundance in the roots was significantly influenced by spike formation. *proteobacteria* was the dominant phylum in the roots (47–67%), and therefore, most of its significantly enriched gene subgroups were more abundant than those of *Actinobacteria* and *Bacteroidetes*, even when calculating abundance per cell (Appendix A). 

## 3. Discussion

Roots select their associated microbiome members and affect their activity by providing a nutrient-rich environment based on distance from the root [29]. Some of the rhizodepositions subsequently diffuse from the root surface into the rhizosphere [30]. Many crops, including wheat, have been found to secrete increasing amounts of organic carbon for the first 50 days after planting, after which organic carbon deposition decreases dramatically [21]. From that point on, root senescence is occurring rapidly and eventually resulting in root decomposition. We propose, based on metagenomic analyses, that root-associated and rhizosphere community abundance, structure and their gene repertoires respond to a turning point in wheat development (spike formation) that occurs concomitantly with changes in root organic carbon depositions.

In this study, in field grown wheat, root-associated bacteria abundance per root increased after spike formation. This increase in bacteria abundance is mainly attributed to decomposition of wheat roots at the final stages of wheat growth manifested by a decrease in TEF gene copy number while total bacteria abundance remains comparable through weeks 12, 17 green and 17 yellow. 

Applying two genomic approaches (DNA sequencing and qPCR) we demonstrated that there is a potential for higher abundance of genes related to denitrification in rhizosphere compared to root-associated microbiome during wheat growth, possibly due to competition with the plant over nitrogen in the root vicinity. Based on the abundance of urea transporter genes, root-associated bacteria possibly import urea, used as a fertilizer in the field, as a source of nitrogen for building amino acids and nucleotides.

Spatial-temporal dynamics of root development leads to a constant need for bacteria to follow the root and re-assemble the associated microbiome [31,32]. Root-associated bacteria have important traits that provide them with a selective advantage in colonization. These include, among others, attachment and competition mechanisms such as motility and chemotaxis, that enable sensing and reaching the root surface [33,34]. It has been shown that non-motile or reduced-motility mutants are unable to compete in root colonization [35]. Another trait is biofilm formation, which creates a better protected niche [36]. In this study, three microbiome functional clusters were shown to be specifically associated with root proximity: biofilm and sensorial movement; antibiotic production and resistance; carbon biosynthesis, degradation and transporters, and amino acid biosynthesis and transporters. The first cluster includes genes related to secretion systems (type I, II, III, IV and VI, pili, flagella and MSHA), biofilm formation, motility, quorum-sensing and chemotaxis. In the second cluster, half of the genes were associated with biosynthesis of polyketide antibiotics which are known to facilitate competition and communication between microorganisms [37]. The third gene cluster included transporters for carbon and amino acids which were significantly enriched in the roots compared to the rhizosphere. Transporters of fructose, glycerol and glucose and mannose were enriched in the root-associated bacteria compared to the rhizosphere. Indeed, those compounds have been recently identified as wheat root exudates [38]. Based on root-associated bacterial gene enrichment, it may be concluded that during root colonization and growth, bacteria sense the presence of emerging roots and move toward them from the bulk soil, express resistance genes to compete with other bacteria, then utilize the exudates, proliferate on the root surface and form a biofilm.

Interestingly, in the root-associated microbiome, the three functional gene clusters were negatively influenced at the time of spike formation. The biofilm and sensorial movement genes were abundant before spike formation, while the roots are growing and bacterial colonization is actively occurring, but none of these genes were abundant after spike formation. Moreover, genes related to antibiotic resistance mechanisms were abundant before spike formation. These include genes coding for resistance to beta-lactams and antimicrobial peptides, both provide tools to remove competing organisms, as well as CRISPR/Cmr systems that provide immune responses directed against the RNA of invading elements [39,40]. However, after spike formation, genes related to the biosynthesis of various antibiotics, including polyketides, were enriched. This observation suggests that during colonization and biofilm formation, bacteria need genes that will allow them to cope with antibiotics producers, whereas in the mature biofilm stage, they invest in inhibiting new colonizers. 

Before and after spike formation, a large portion of significantly abundant genes were related to carbon and amino acid transporters, especially ABC transporters. Amino acids, important component of root exudates, are crucial for the early steps of root colonization [41] and indeed, more than twice the amount of amino acid-related genes were abundant in the root microbiome before as compared to after spike formation. Changes in gene abundance of those bacterial carbon and amino acid transporters may probably reflect the carbohydrate and amino acid quantities and types of exudates produced at different stages of plant development. For instance, glucose and mannose, glycerol, inositol, methyl galactoside and trehalose are known wheat root exudates [38], and indeed their transporters were enriched in the microbiome before spike formation. However, transporters for fructose, sorbitol and mannitol, also known wheat root exudates [38], were abundant in the root microbiome after spike formation. Chemical assessment of the actual composition of root exudates in the soil is difficult and inaccurate [42], thus, it is suggested that microbiome-abundant transporter-encoding genes may reflect the composition of the root exudates available to the bacteria in situ.

Nine functions whose gene abundance was significantly influenced by spike formation in both root-associated and rhizosphere niches. Genes related to these functions sum up to half of the total significantly enriched genes in the rhizosphere. Moreover, all of these genes are enriched in the rhizosphere before spike formation, when root exudation is expected to be at its peak [21], supporting the notion that root exudates diffuse into the surrounding soil, and contribute to creating the rhizosphere.

In this study, the wheat root community consisted mostly of *proteobacteria, Actinobacteria* and *Bacteroidetes,* known to be wheat root colonizers [43,44,45]. Interestingly, in the root-associated niche, the abundance of *proteobacteria* and *Actinobacteria* was influenced by spike formation, whereas that of *Bacteroidetes* was not. *Proteobacteria* were significantly more abundant and, per cell, had a higher abundance of significantly enriched genes before spike formation (i.e., genes related to carbon metabolism and transporters, amino acid metabolism, degradation and transporters, antibiotic resistance, quorum-sensing, chemotaxis, biofilm formation, and various types of secretion systems, denitrification and metal and ion transporters) compared to *Actinobacteria.* On the other hand, *Actinobacteria,* which were significantly more abundant after spike formation, had more significantly enriched genes at this time point per cell (i.e., genes related to carbon transporters, degradation and metabolism, amino acid transporters and biosynthesis) compared to *proteobacteria.* Some *Actinobacteria* have the ability to partially degrade plant cell wall components, such as lignocellulose and other complex molecules [46], which are characteristic of mature and remnants of wheat roots [47]. In addition, genes encoding cellobiose transporter were abundant after spike formation, suggesting consumption of cellobiose, a product of cellulose hydrolysis [48]. These changes in abundance with spike formation may indicate a major role for *proteobacteria* and *Actinobacteria* in wheat root colonization by adapting their functional profile to follow root growth and secretion patterns, as well as decay. 

We demonstrated functional and phylogenetic division in the microbiome of wheat root zone in both time and space (pre- and post-spike formation, and root-associated vs. rhizospheric niches). As it is not feasible to directly examine mechanisms occurring underground under field conditions, we suggest and discuss several possible plant-microbe interactions based on traits found in the metagenome. Differences in wheat bacterial community structure and gene functions manifest the functional division between niches as well as developmental stages, reflecting plant physiology status during plant and its microbiome co-development in the field. These findings shed light on the dynamics of the fascinating plant–microbe and microbe–microbe interactions in the root zone.

## 4. Materials and Methods

### 4.1. Soil and Root Sampling

Wheat (*Triticum aestivum* cv. Tishrey) was cultivated in a field at the Volcani Center, Rishon LeZion, Israel. The soil was clay loam (36% clay, 22% silt and 42% sand) and the soil parameters prior to wheat sowing were: pH 7.99 ± 0.04, electrical conductivity 99 ± 2 µS/m, N-NO_3_ 0.55 ± 0.05 mg/kg, NH_4_ 0.36 ± 0.02 mg/kg, P-PO_4_ 0.06 ± 0.01 mg/kg, total soluble organic carbon 4.64 ± 0.08 mg/kg, and total soluble nitrogen 0.87 ± 0.01 mg/kg. The field was amended once with cattle urine (48 m^3^/acre) prior to wheat sowing in November 2015. Weather conditions during wheat growth are provided in Appendix A.

During wheat growth and development, microbiomes were sampled as follows: rhizosphere microbiome was collected from the soil dislodged from the roots by shaking. The samples were immediately placed on ice and stored at −80 °C for further analyses. For the root-associated microbiome, roots were washed three times with saline solution (0.85% NaCl) and immediately placed on ice and stored at −80 °C for further analyses. Samples were collected at the following developmental stages: emergence with one or two leaves (1st week, 16 December 2015), tillering (2nd week), jointing (5th week), heading (9th week), anthesis (12th week) and spike maturation (17th week). Root and rhizosphere samples were collected in triplicate in weeks 1, 2, 5, 9 and 12; in week 17, two subsamples were collected for each spike category (i.e., green and mature yellow, Appendix A).

### 4.2. Rhizosphere and Root DNA Extraction for Sequencing and qPCR 

To obtain the root-associated microbiome, wheat roots were vortexed three time with 85% saline solution, until no visible soil particles were attached to the roots. Total DNA was extracted from 0.4 g of complete root system and 0.3 g rhizosphere soil, using the Exgene Soil DNA mini-isolation kit (GeneAll, Seoul, Korea) according to the manufacturer’s instructions.

### 4.3. Generation of qPCR Plasmid Standards

Plasmids containing the 16S rRNA gene, translation elongation factor 1 (*TEF*, a plant housekeeping gene) and *nosZ* gene were generated as described previously [49,50,51]. Each PCR amplification product was ligated into pGEM-T Easy Vector (Promega, Madison, WI, USA) and plasmids were transformed into BioSuper *Escherichia coli* DH5α competent cells (Bio-Lab, Jerusalem, Israel). Circular plasmid DNAs were used as the standards to create calibration curves at 10-fold dilutions for gene quantification by real-time qPCR.

### 4.4. Assessment of Gene Copy Numbers by qPCR 

Copy numbers of total bacterial community (16S rRNA gene), plant housekeeping gene (*TEF*) and selected denitrifying gene (*nosZ*) were assessed using selected primers (Appendix A) in wheat roots using the StepOnePlus Real-Time PCR System (Applied Biosystems, Foster City, CA, USA). Triplicates from whole genomic DNA were diluted up to 6 ng/µL and 1 µL was used in a 20-µL final reaction volume together with 50 µM forward and revers primers and 10 µL 1X FAST or Power SYBR MasterMix (Thermo Fisher Scientific, Waltham, MA, USA). Three biological and three technical replicates were analyzed for each individual soil or root DNA sample. Reaction efficiency was monitored in each run by means of an internal standard curve (constructed plasmids) using duplicates of 10-fold dilutions of standards ranging from 10^8^–10^2^ copies per reaction. Efficiency was 89–95% for all target genes and runs, and R^2^ values were greater than 0.99. Copy numbers of the target genes were calculated based on the relative calibration curve of plasmid copy numbers. All data analyses were conducted using StepOne software v2.3 (Applied Biosystems, Waltham, MA, USA). 

### 4.5. Shotgun Sequencing

Shotgun metagenomic libraries were prepared as described previously [52]. The final size-selected pool was sequenced in an Illumina NovaSeq instrument with an S4 flow cell, employing 2 × 150 base reads. Library preparation and pooling were performed at the University of Illinois at Chicago Sequencing Core (UICSQC), and sequencing was performed by Novogene Corporation (Chula Vista, CA, USA).

In total, we obtained 1019 Gb of information, with 33–44 million sequences per root sample and 34–53 million sequences per rhizosphere sample. These sequence data were submitted to the Sequence Read Archive (SRA) of the NCBI databases under accession number SUB8180046, BioProject: PRJNA664890.

All reads were subjected to quality control using FastQC v0.11.3 [53] and barcode trimming using Trimmomatics v0.32 [54]. Reads were mapped to the whole wheat metagenome using Bowtie2 v2.3.5.1 [55], and mapped reads (wheat genes) were filtered out from each sample. Then, short Illumina reads in triplicate from each week (1, 2, 5, 9 and 12) were assembled using SPADES v3.13.0 [56] into longer contigs, to create five wheat-root microbiome catalogues, one for each week. Six rhizosphere microbiome catalogues were created, one for each week (1, 2, 5 and 9), and two for week 12 due to its overall size. Reads with four replicates from week 17 were separated into two root microbiome catalogues and two rhizosphere microbiome catalogues. Altogether seven root-associated catalogues and eight rhizosphere catalogues were created (Appendix A). Prodigal v2.6.2 [57] was used for protein-coding gene prediction for each of those 15 catalogues. To create a non-redundant set of genes for each niche separately, we used CD-HIT-EST software v4.8.1 [58] with a similarity threshold of 95%. Those genes were used as the root-associated gene catalogue, which included 53 million partial genes, and for the rhizosphere gene catalogue, which included 91 million partial genes. These gene catalogues were searched against the non-redundant NCBI protein database using DIAMOND sensitive algorithm v0.9.24.125 [59] to assign taxonomic and functional annotations. Results were then uploaded to MEGAN Ultimate edition software v6.15.2 [60]. The lowest common ancestor (LCA) algorithm was applied (parameters used with minimum bit score of 70, minimum support of 5%, and 30% top threshold) to compute the assignment of genes to specific taxa. For functional annotation, the Kyoto Encyclopedia of Genes and Genomes (KEGG) database [61] was used. Following annotation, to generate taxonomic and functional count tables, each library was mapped to the gene catalogue with Trinity mapping software v2.8.4 [62], with Bowtie2-modified parameters (--no-unal --gbar 99999999 -k 250 --dpad 0 --mp 1,1 --np 1 --score-min L,0,-0.9 -L 20 -i S,1,0.50). 

### 4.6. Data Analyses

All sequencing data analyses were performed using R statistical software. A non-metric multidimensional scaling (nMDS) ordination plot was constructed using R package VEGAN v.2.5-5 [63]. The data matrix was transformed using normalized count transformation. For community structure, ordination was generated using the Bray–Curtis dissimilarity matrix, and for community function of KEGG orthologous (KO) groups, ordination was generated using the Euclidean dissimilarity matrix. Permutational multivariate analysis was used to compute the variance between microbial community structure or function and experimental parameters (niche and spike), using the Adonis function in the Vegan R package [64]. Changes in structural and functional similarity between wheat root and rhizosphere microbial communities in each week were calculated using Student’s *t*-test in JMP 14 Pro software (SAS Institute Inc., Cary, NC, USA) and statistical significance was set at *p* < 0.05. Network co-occurrence analysis of bacterial communities in root-associated and rhizosphere samples were constructed using Cytoscape v.3.7.2 software’s [65] CoNet app [66]. For comparison of taxonomic changes and functional traits, differential abundance was estimated using DESeq2 [67] and was considered significant when the difference in abundance between genes had a false discovery rate (FDR)-adjusted *p*-value <0.05. For comparison of changes in taxonomic and functional traits, the bacterial read counts table was binned into KO groups, based on DIAMOND-MEGAN annotation. 

## Figures and Tables

**Figure 1 ijms-22-11948-f001:**
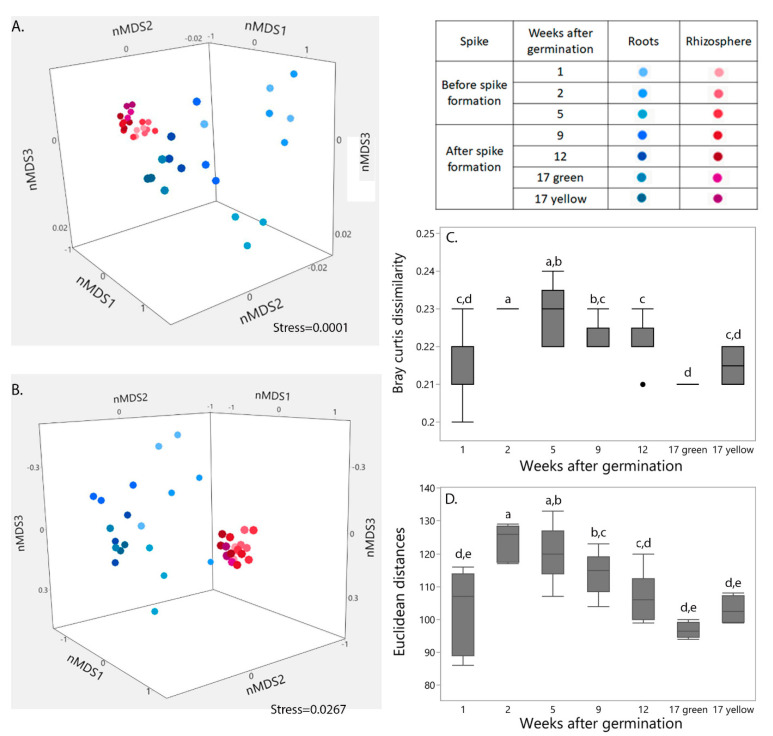
Shifts in root-associated and rhizosphere bacterial community structure and function with plant age. nMDS ordination plots show clustering patterns of wheat root-associated and rhizosphere microbial community structures (**A**) and functions (**B**) during wheat growth. Data matrix was transformed using normalized count transformation in the DESeq2 package, and then ordination was generated using Bray-Curtis dissimilarity and Euclidean distances, respectively. (**C**) Changes in structural and (**D**) functional similarity between wheat root-associated and rhizosphere microbial communities at each week. Different lowercase letters indicate significant difference (*p* < 0.05) by Student’s *t*-test. Weeks 1, 2, 5, 9 and 12: *n* = 6; week 17 green (green spike) and yellow (yellow spike): *n* = 2 for each. A black dot in Figure 1C, indicate the outlier at week 12.

**Figure 2 ijms-22-11948-f002:**
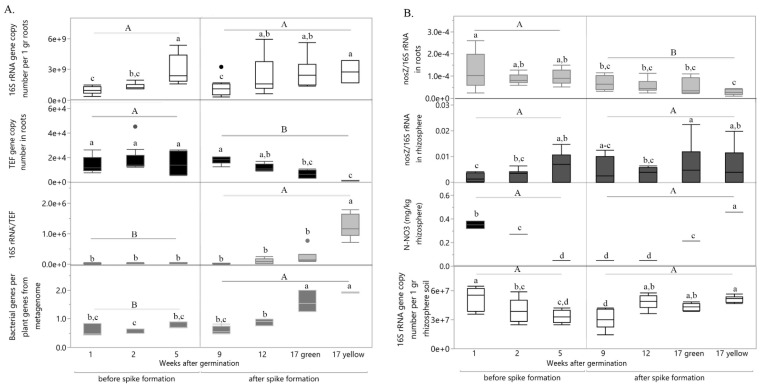
Influence of wheat developmental stage on total bacterial community and *nosZ* gene abundance in roots and rhizosphere. (**A**) Changes in abundance of total bacterial community (16S rRNA) and its relative abundance per root tissue (e.g., per the plant housekeeping gene *TEF*, and per total plant genes from metagenome) during wheat growth. (**B**) Changes in relative abundance of denitrifying gene *nosZ* per total bacterial community in roots and rhizosphere as well as nitrate content in rhizosphere during wheat growth. Each bar represents the average of three biological each of three technical replicates with standard error. Different uppercase and lowercase letters indicate significant difference (*p* ≤ 0.05) by Student’s t test. N-NO_3_ = nitrate nitrogen.

**Figure 3 ijms-22-11948-f003:**
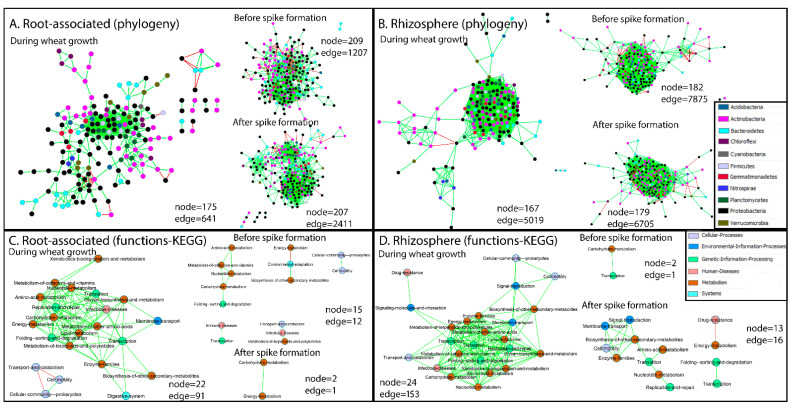
Network co-occurrence analysis of bacterial communities of root and rhizosphere samples constructed using CoNet, a Cytoscape plugin. Green and red links (i.e., edges) indicate significant positive and negative correlations between two nodes, respectively. In (**A**,**B**), each node represents taxa affiliated to all taxonomic levels from the metagenome, labeled at the phylum level. In (**C**,**D**), each node represents a group of functional genes, annotated with known KO groups from the KEGG database.

**Figure 4 ijms-22-11948-f004:**
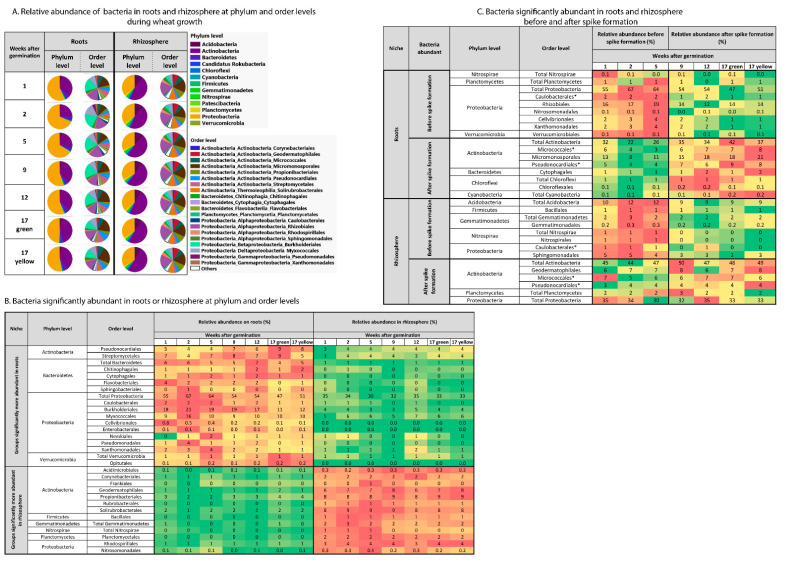
Changes in root-associated and rhizosphere bacterial community structure as a function of niche (roots vs. rhizosphere) and plant age. (**A**) Shifts in root and rhizosphere bacterial community structure at the phylum and order levels as influenced by plant age. Significantly abundant groups in root or rhizosphere (niche) (**B**) and with spike formation (**C**). Changes in bacterial abundance were calculated using DESeq2 with cutoff FDR-adjusted *p*-value < 0.05. Before spike formation, *n* = 9; after spike formation, *n* = 10. Asterisk (*) indicate bacteria groups at order level, that their abundance in root and rhizosphere microbiome was influence by spike formation.

**Figure 5 ijms-22-11948-f005:**
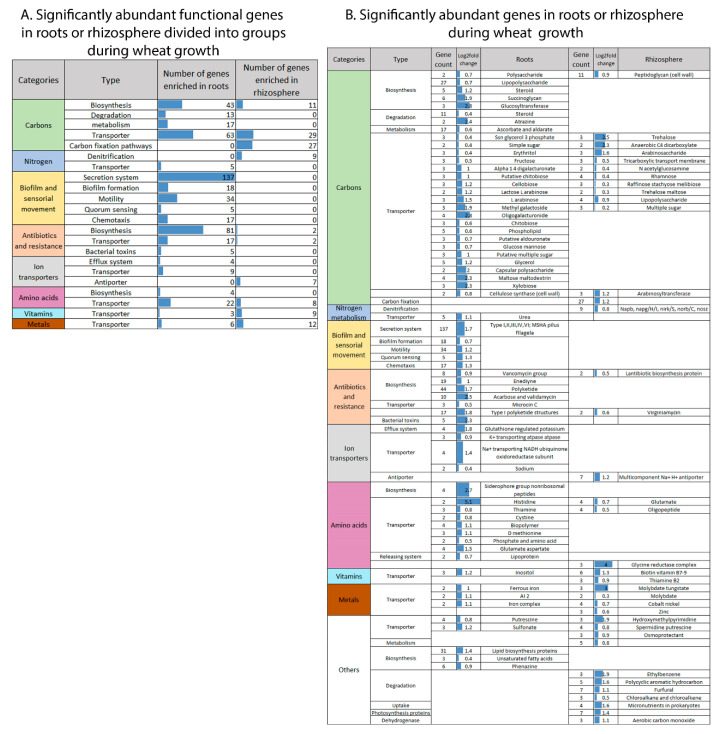
Spatial effect on microbiome functions. Significantly abundant functional genes in two niches (root-associated vs. rhizosphere) summarized in functional groups (**A**) and genes encoding for those functional groups (**B**).

**Figure 6 ijms-22-11948-f006:**
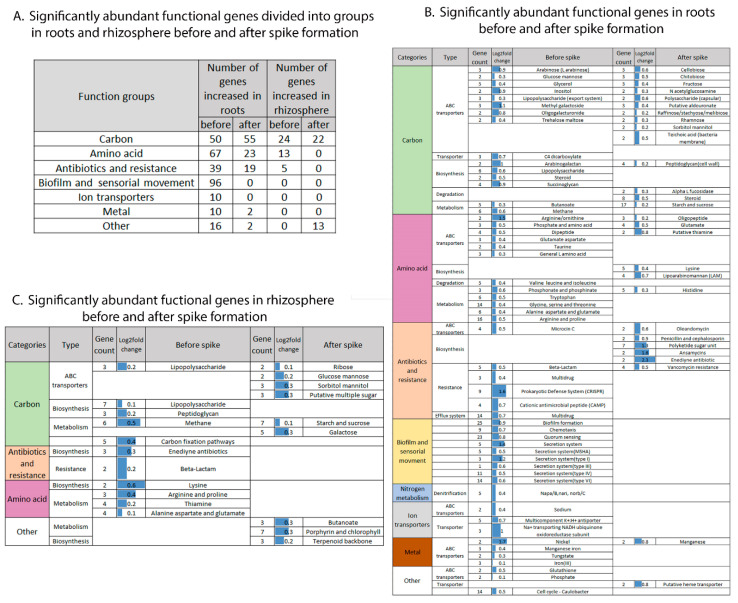
Temporal effect on microbiome functions. Significantly abundant functional genes before and after spike formation, divided into groups (**A**) and in roots (**B**) and rhizosphere (**C**).

**Figure 7 ijms-22-11948-f007:**
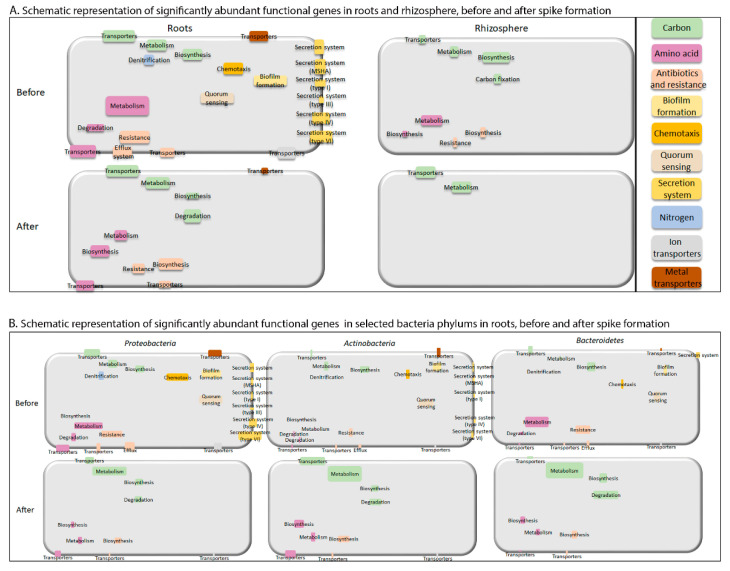
Schematic model presenting significantly abundant functional genes. (**A**) Schematic representation of significantly abundant functional genes before and after spike formation in roots and rhizosphere. Size of each rectangle is proportional to sum of significantly enriched functional gene counts in each subcategory, calculated based on data in Figure 6B,C. (**B**) Schematic representation of significantly abundant functional genes before and after spike formation in roots in the three selected, dominant bacterial phyla *proteobacteria*, *Actinobacteria* and *Bacteroidetes*. Size of each rectangle is proportional to normalized amount of the gene functions (using normalized count transformation in DESeq2 package), divided by relative abundance of each relevant taxa. List of all normalized counts is provided in Appendix A.

## Data Availability

The sequence data from root and rhizosphere samples was submitted to the Sequence Read Archive (SRA) of the NCBI databases under accession number SUB8180046, BioProject: PRJNA664890.

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
