# Peer review of "Spike Formation Is a Turning Point Determining Wheat Root Microbiome Abundance, Structures and Functions"

_ijms, 2021, doi:10.3390/ijms222111948_

Round 1

Reviewer 1 Report

ijms-1432936

In the present study “Spike formation is a turning point determining wheat root mi-crobiome abundance, structures and functions”, authors describe the changes in root-associated and rhizosphere bacterial community structure and functions with stages of plant development. I think that the work falls into the scope of the journal and findings are interesting, however MS demands minor revision.

Comments:

Abstract: This section seems scary and unclear. I would suggest to explain your results in detail. 

Introduction: The significant novel point of the study over the precedent studies is not clear. There are some old references cited in this section that needs to be replaced with ones.

Materials and methods: Weather data maybe need?

Results and Discussion: In results, there is a striking lack of connectors between sentences and leading to confusing. One way of improving Discussion is to avoid repetition of results in this part. Discussion is very shallow and need in depth discussion with the recent literature published.

Kind regards!

Author Response

Our response to reviewers comments are in red:

Abstract: This section seems scary and unclear. I would suggest to explain your results in detail. 

Additional results were included and explained. Please see the attachment.

Introduction: The significant novel point of the study over the precedent studies is not clear. There are some old references cited in this section that needs to be replaced with ones.

Recent references were added to the introduction according to reviewer's suggestion.

Materials and methods: Weather data maybe need?

Weather conditions were provided in Supplemental Fig.1.

Results and Discussion: In results, there is a striking lack of connectors between sentences and leading to confusing. One way of improving Discussion is to avoid repetition of results in this part. Discussion is very shallow and need in depth discussion with the recent literature published.

Repetition of results was deleted from the discussion section and recent references were added to the discussion according to the reviewer's suggestion.

Reviewer 2 Report

The rhizosphere is a zone of biological activity between plant roots and soil harboring a plethora of microorganisms. The key interactions among a multitude of microorganisms in the rhizosphere have a direct or indirect effect on the plant. Being versatile and intriguingly complex, a comprehension regarding the elementary principles of microbial ecology and functioning is significant to enhance the plant productivity and agroecosystem working. The interplay between plant roots and the associated microbes is regulated by profound chemical signaling. Most of the known facts about these interactions till recently have been derived through the studies based on culturing the microbes; however, it is an established fact that majority of the microbes are uncultivable. Novel insights into enhancing our ability to unravel the quintessential factors determining the rhizosphere microbiome could offer the progress towards the development of sustainable agriculture. We now have the opportunity to utilize the advanced culture independent techniques to have an insight into the intriguing plant-microbe interplay.

In the present work, the authors investigate the metagenomic analysis of rhizosphere microbiome in pre- and post-spike formation time points. To  understand  plant-soil  functioning,  it  is  necessary  to  model  the  distribution patterns and functional gene repertoires of soil microorganisms. Metagenomic analysis combined with functional co-occurrence networks revealed a significant impact of plant developmental stage on its microbiome during the transition from vegetative growth to spike formation (gene functions related to biofilm and  sensorial  movement,  antibiotic  production  and  resistance,  and  carbons  and  amino  acids  and their transporters). This part of the manuscript is interesting and clearly provides new data valuable for the research community. The Authors’ metagenomic studies present a strong mandate to understand the enormous richness and diversity of rhizosphere microbiome as well as the key biological processes. The authors have done an excellent job at describing the problem, the methods and the results.

GENERAL COMMENTS:

TITLE

The paper title is well stated, it is informative and concise.

ABSTRACT, INTRODUCTION

Abstract and Introduction were well written.

MATERIAL AND METHODS

Material and research methods are presented appropriately and clearly. Experimental setup and the description in the methods section are well structured, and the statistical analysis is done alright.

RESULTS

The results obtained in this study are interesting. Results presented correctly.

DISCUSSION

In general, the discussion of results is correct and sufficient.

LITERATURE

The items of literature included in the paper are rather sufficient and adequate to the subject of the paper.

The text of the manuscript is not formatted correctly yet.

Author Response

Our response to reviewers comments are in red:

TITLE
The paper title is well stated, it is informative and concise. 

Thanks 

ABSTRACT, INTRODUCTION
Abstract and Introduction were well written.

Thanks 

MATERIAL AND METHODS
Material and research methods are presented appropriately and clearly. Experimental setup and the description in the methods section are well structured, and the statistical analysis is done alright.

Thanks 

RESULTS
The results obtained in this study are interesting. Results presented correctly. 

Thanks 

DISCUSSION
In general, the discussion of results is correct and sufficient.

Thanks 

LITERATURE
The items of literature included in the paper are rather sufficient and adequate to the subject of the paper.

Thanks 

The text of the manusctipt is not formatted correctly yet.
